# NR4A1 Acts as a Nutrient Sensor That Inhibits the Effects of Aging

**DOI:** 10.3390/nu17162709

**Published:** 2025-08-21

**Authors:** Stephen Safe

**Affiliations:** Department of Veterinary Physiology and Pharmacology, College of Veterinary Medicine, Texas A&M University, College Station, TX 77843, USA; ssafe@cvm.tamu.edu; Tel.: +1-(979)-845-5988; Fax: +1-(979)-862-4929

**Keywords:** NR4A1, aging, health-protective receptor, polyphenolic target

## Abstract

Orphan nuclear receptor 4A1 (NR4A1) is a member of the NR4A subfamily that was initially discovered as an intermediate early gene expressed in response to stressors, including inflammatory agents. This review addresses the hypothesis that NR4A1 is a key nutrient sensor that contributes to the anti-aging and health-protective effects of receptor ligands, dietary phenolics, and other diet-derived compounds. There is evidence in animal models including humans that NR4A1 serves as an important gene that decreases the rate of aging and its associated diseases. For example, in humans and mice, NR4A1 expression decreases with age and loss of NR4A1 enhances disease susceptibility, and survival curves show that NR4A1-deficient mice live 4 months less than wild-type animals. An extensive comparison of inflammatory diseases, immune dysfunction, and fibrosis in multiple tissues shows that in NR4A1^−/−^ mice and rats these diseases and injuries are enhanced compared to wild-type NR4A1^−/−^ animals. There is evidence showing that structurally diverse NR4A1 ligands reverse the induced adverse effects in NR4A1 wild-type mice. This raises an important question regarding the mechanisms of NR4A1-dependent inhibition of the aging process and the potential for this receptor as a nutrient sensor. It has been well established that polyphenolics, including flavonoids, resveratrol, and other compounds in the diet, are health-protective and decrease the aging process. Recent studies show that resveratrol and flavonoids such as quercetin and kaempferol bind NR4A1 and exhibit protective NR4A1-dependent inhibition of endometriosis and cancer. These limited studies support a role for NR4A1 as a potential dietary sensor of nutrients that are known to be health-protective and a potential nutrient target for improving health.

## 1. Introduction

The rate of aging in individuals is complex and is influenced by multiple factors, including genetics, environmental factors, diet, and lifestyle [1,2,3,4]. The goal of decreasing mortality by addressing genetic deficiencies is difficult and requires heightened awareness and early treatment of specific genetic-induced diseases. Increased health benefits are more attainable by modifying diet/lifestyle and exposure to adverse environmental factors. Diet is an important consideration, since population studies show decreased rates of mortality in vegetarians and individuals consuming the Mediterranean diet and other diets enriched in fruits, nuts, and vegetables and also coffee [5,6,7,8,9,10]. These health benefits are primarily due to modulation of key pathways/genes by dietary components, which include flavonoids and other polyphenolics, caloric restriction, vitamins, and microbiome-derived compounds.

Some of the genes/pathways involved in aging include sirtuin-1 (SIRT-1), SIRT-3, SIRT-7, telomerase components, AMPK/mTOR signaling, FOX03, NRF2, NFkB, and polyamine production. The functions of these genes are variable, and they are involved in redox regulation, anti-inflammatory pathways, mTOR pathway inhibition, maintenance of DNA integrity (e.g., DNA repair mechanisms), enhanced mitochondrial functions, and extracellular matrix remodeling [1,2,3,4]. The aging process is accompanied by the decline in many of the important pathways/genes required for maintaining cellular homeostasis, and dietary interventions that decrease mortality modulate the aging process by targeting these specific pathways and related genes. The identities of some genes/pathways involved in aging and many of the dietary compounds that modulate and enhance health benefits such as decreased mortality are known; however, the mechanisms of individual and compound-induced protective responses are not completely understood. Identification of specific pathways/genes targeted by health-promoting dietary compounds would facilitate the development of a more precision nutrition approach for enhancing diet-induced health and the development of related dietary interventions and therapeutics.

We hypothesize that orphan nuclear receptor 4A1 (NR4A1) plays a protective role against stressor/inflammation-induced cell damage that accompanies the aging process. In addition, synthetic NR4A1 ligands further enhance NR4A1-mediated protection, and recent discoveries show that some health-promoting polyphenolics also bind NR4A1, suggesting that this receptor may also play a role as a nutrient sensor.

## 2. Orphan Nuclear Receptor 4A1 (NR4A1) and Aging

### 2.1. Background

Orphan nuclear receptor 4A1 (NR4A1, Nur77) is a member of the NR4A subfamily of nuclear receptors that includes NR4A2 (Nurr1) and NR4A3 (Nor1), and these receptors are classified as orphans since their endogenous ligands have not yet been identified [11,12,13]. NR4A1 and other members of this subfamily have a modular structure that exhibits similar percentages of amino acid identities in the C-terminal ligand-binding (E) and DNA-binding (C) domains but differs substantially in its N-terminal (A/B) domain. NR4A1 targets multiple promoter DNA-binding elements and also modulates gene expression through activation of regulated genes via protein–protein interaction (Figure 1) [14,15,16,17]. The three NR4A members are immediate early genes in non-transformed cells and they are induced by multiple stressors in various cell types [18,19,20,21], whereas in cancer cells individual NR4A genes exhibit tumor type-specific tumor suppressor- or tumor promoter-like activities [14]. Although the endogenous ligands for NR4A1 have not been identified, a large number of biochemicals including prostaglandins and fatty acids bind and activate the receptor at pharmacological concentrations. In addition, dietary compounds, structurally diverse natural products and their analogs, and a large number of synthetic chemicals including drugs also bind NR4A1 and activate the receptor [22,23,24]. Studies in this laboratory have identified and characterized a series of synthetic 1,1-bis(3′-indolyl)-1-(substitutedphenyl)methane (CDIM) compounds derived from 1,1-bis(3′-indolyl)methane (DIM) that bind NR4A, and the 3,5-disubstitutedphenyl (DIM-3,5) analogs bind both NR4A1 and NR4A2 [25] (Figure 2).

### 2.2. Effects of Aging on NR4A1 Expression

A characteristic of many key genes involved in reducing the rate of aging is their decreased expression during the aging process, and this has also been observed for NR4A1. For example, NR4A1 mRNA expression is significantly decreased in peripheral blood mononuclear cells from ages 20–40 to >75 years of age, and similar differences were observed in mice [26]. Moreover, in a subset of humans with normal and decreased cognitive function, NR4A1 expression was lower in the latter group. Experimental studies in mice showed that the age-dependent decrease was in the hippocampal CAI pyramidal neurons and the loss of NR4A1 and impairment of cognition was due to loss of Trk, an NR4A1-regulated gene [26]. Chronic kidney disease is age-related and associated with enhanced tubulointerstitial fibrosis; in mice there was an inverse correlation between TGFβ expression vs. NR4A1 expression and increased kidney damage. In older mice, the loss of NR4A1 activated TGFβ/SMAD signaling and subsequent renal toxicity [27]. The age-dependent decrease in NR4A1 was also observed in the mouse liver, kidney, and peri-adipose tissue; this was paralleled by a decrease in SIRT1 expression and was also observed in oxidative stress-induced HEK-293T cells [28]. NR4A1 indirectly regulates SIRT1 by decreasing the E3 ligase MDM2 and thereby enhancing SIRT1 expression by decreasing its proteasome-mediated degradation. This paper also compared the survival of wild-type and NR4A1^−/−^ mice and showed that the loss of NR4A1 decreased % survival curves of mice by approximately 4 months [28]. Thus the mouse models showed that NR4A1 is a pro-survival transcription factor due in part to stabilizing SIRT1.

The extent of cardiac fibrosis is also age-dependent and comparisons between 6- and 15-month-old mice showed increases in cardiac fibrosis and damage that correlated with decreased expression of NR4A1 mRNA levels [29]. Moreover, in NR4A1^−/−^ mice the markers of cardiac dysfunction were enhanced compared to wild-type mice. Mechanistic studies showed that NR4A1 deficiency correlated with decreased GSK-3β and increased β-catenin expression, and the former gene was a direct target of NR4A1. These results also link NR4A1 activity to SIRT3, which inhibits age-related fibrosis in the heart and other tissues by SIRT3-dependent deacetylation and activation of GSK-3β [30]. The contributions of NR4A1 to muscle growth were also investigated in wild-type and NR4A^−/−^ mice at various time points [31]. Although age-dependent changes in NR4A1 were not provided, the loss of NR4A1 (globally and muscle-specific) decreased myofiber size in mice at E 18.5 and in 3-month-old mice. Moreover, the loss of NR4A1 was accompanied by induction of two negative regulators of muscle mass, SMAD2 and FOX01. Ovarian function also decreases in reproductively aging mice, and this is accompanied by decreased NR4A1 expression. Moreover, in human umbilical cord mesenchymal stem cell transplantation models in mice, AMPK/NR4A1 signaling enhanced ovarian function and decreased fibrosis [32,33,34]. Results of these age-dependent effects on expression of NR4A1 and mechanistic studies with NR4A1^−/−^ mice demonstrate that like other pro-survival genes NR4A1 decreases with age and NR4A1 contributes to several health benefits. These studies also show that NR4A1 interacts with genes that are associated with decreased rates of mortality, including SIRT1, GSK-3β, and SIRT3. Moreover, NR4A1 also plays a protective role in maintaining DNA integrity. DNA-dependent protein kinase (DNA-PK) plays an important role in double-stranded break (DSB) repair and NR4A1 is involved in repair of DNA damage [35,36]. NR4A1 and other NR4As are recruited to the DNA damage sites and target poly-ADP-ribosylated DNA-PKs to enhance its autophosphorylation.

Thus the role of NR4A1 in aging shows that this transcription factor interacts with several key genes involved aging (Figure 3), including SIRT1 in oxidative stress responses [28], SIRT3 in age-related fibrosis [30], and NFkB in multiple stress-induced responses [18], DNA damage, and repair responses [35,36]. In addition, NR4A1 regulates mTOR signaling in angiotensinαII-induced cardiac hypertrophy [37]; NR4A1 also enhances muscle mass in mice through IGF1-induced activation of mTOR [38], and in a mouse model of Parkinson’s disease the anti-inflammatory effect of NR4A1 is also associated with NRF2 [39]. FOX03 expression inhibits mouse hepatocyte proliferation, and this is paralleled by downregulation of NR4A1 in a liver regeneration model [40]. Thus NR4A1 regulates or exhibits some association with many of the genes that play a role in aging; however, their interactions and responses and effects of stressors may be variable and cell context-dependent. The next section of this review briefly outlines several tissue-specific health-promoting functions of NR4A1, particularly with respect to protecting against stressor/inflammation-induced damage, which plays a role in the aging process. We hypothesize that orphan nuclear receptor 4A1 (NR4A1) plays a protective role against stressor/inflammation-induced cell damage that accompanies the aging process. In addition, synthetic NR4A1 ligands further enhance NR4A1-mediated protection, and recent discoveries show that some health-promoting polyphenolics also bind NR4A1, suggesting that this receptor may play a role as a nutrient sensor.

## 3. Tissue-Specific Functions of NR4A1

Although there are only a limited number of studies on the role of NR4A1 in aging, it has been shown that NR4A1 decreases with age, and NR4A1 deficiencies are accompanied by enhanced adverse responses. There is an increasing number of publications on the tissue-specific constitutive role of NR4A1 and these are primarily determined by examining responses after knockdown of the receptor, and this will be outlined in this section.

### 3.1. Cardiovascular

NR4A1 is expressed in the heart and is induced by VEGF and other vascular permeabilizing agents [41,42]. In mice, the loss of NR4A1 is accompanied by lower levels of basal vascular permeability (BVP), and it was concluded that NR4A1 plays a key role in BVP, chronic vascular hyperpermeability (CVH), and acute vascular hyperpermeability (AVH) [42,43]. In NR4A1-KO mice adverse cardio remodeling was associated with increased diastolic and systolic Ca^+2^ and this was accompanied by larger cardiomyocytes [44]. Protection of cardiac remodeling by NR4A1 was also due to NR4A1-dependent suppression of the sympathetic co-transmitter neuropeptide Y (NPY) [45]. NR4A1 also plays an important role in immune system-mediated cardiac remodeling, and this involves Ly-6C^high^ monocytes and macrophages that infiltrate the damaged myocardium [46,47,48]. The loss of NR4A1 in macrophages and monocytes results in enhanced pro-inflammatory M1 macrophages and NR4A1-KO mice maintained on a Western diet developed increased atherosclerosis [47]. These results clearly demonstrate a protective role for NR4A1 in cardiovascular disease and there are also extensive studies on interactions of NR4A1 during induced cardiovascular damage [49], which will also be noted in the next section of this review.

### 3.2. Neuronal

NR4A1, NR4A2, and NR4A3 play an important role in long-term memory and NR4A ligands reactivate age-dependent memory decline [50,51,52] and this is associated with their regulation of endoplasmic reticulum chaperones [53]. Another study in mice reported that NR4A1 was necessary for object location whereas NR4A2 was required for long-term memory, object location, and recognition [54] and loss of NR4A1 in inhibitory GABAergic interneurons affects associative learning [55]. NR4A1 is also involved in post-stroke recovery, and deletion of NR4A1 in microglia results in increased expression of TNF and this results in increased brain injury [56].

### 3.3. Muscle Mass and Myofiber Size

NR4A1 overexpression in the skeletal muscle of mice increased muscle mass and loss of NR4A1 reduced muscle mass and myofiber size via activating IGF-1 growth-promoting pathways [38], and estrogen induces NR4A1 in skeletal muscle [57]. NR4A1 overexpression also promoted cell adhesion and fusion in myoblasts by regulating ZEB1 transcription [58].

### 3.4. Retina and Eye

In a genomic screen for transcriptional targets of circadian melatonin and dopamine signaling in mice, NR4A1 was identified as a candidate gene that regulated neurohormone release and functional adaptation and healthiness of the retinal and photoreceptor cells [59].

### 3.5. Obesity and Metabolic Disease

β-Cells play a critical role in insulin secretion and decreases in β-cell function are associated with development of type 1 and type 2 diabetes. Overexpression of NR4A1 (and NR4A3) enhances β-cell expansion, mitochondrial respiration, and insulin secretion; these effects are reversed in NR4A1-KO mice [60,61] and this complements other studies demonstrating a role for NR4A1 in mitochondrial function [62,63,64].

NR4A1 also plays an important role in muscle cells, which are prime sites of glucose metabolism and overall insulin sensitivity and obesity. For example, results of receptor knockdown in muscle cells show that NR4A1 is linked to genes that regulate lipid homeostasis, mitochondrial function, energy expenditure, and glucose metabolism in muscle cells and in vivo [62,63,64,65,66,67,68,69]. In the liver, NR4A1 is involved in gluconeogenesis and increased blood glucose levels, and deletion of the receptor enhances hepatic steatosis and increases expression of lipogenic genes [63,67]. NR4A1 and other NR4A members also play a role in lipid metabolism and are induced by multiple stimuli; however, overall NR4A1 expression inhibits adipogenesis and this is associated with inhibition of mitotic clonal expansion of adipocytes [68]. The overall mechanisms of NR4A1 action in adipose tissue includes NR4A1-mediated repression of PPARƔ2 in white adipose tissue [69], interactions between NR4A1 and STAT3 acetylation, and leptin sensitivity [70]. Moreover, there is also a report showing that female, but not male, NR4A1-deficient mice exhibit increased susceptibility to obesity when maintained on a high-fat diet [71].

### 3.6. Intestinal Inflammation

NR4A1 also plays a role in intestinal inflammatory conditions, which include colitis and inflammatory bowel disease. In NR4A1-deficient mice, elevated expression of genes related to extracellular matrix (ECM) production, metabolism, and cell proliferation were observed in intestinal smooth muscle cells [72]. Another study showed that loss of NR4A1 also increased intestinal mucosal ECM content and markers of cell proliferation and increased α-smooth muscle actin and collagen levels [73]. Similar protective effects of NR4A1 were observed in another study, and it was also reported that NR4A1 SNPs with lower activity than wild-type NR4A1 were associated with increased risks for ulcerative colitis and Crohn’s disease [74]. NR4A1 indirectly inhibited NFkB signaling by negatively regulating TLR-IL1R signaling and preventing TRAF6 deubiquitination [74]. These studies clearly demonstrate that NR4A1 expression inhibits both basal and dextran sodium sulfate-induced intestinal inflammation and subsequent damage.

### 3.7. Wound Healing and Angiogenesis

Wound healing is a critical step in resolving both basal and induced lesions and both gene knockdown and overexpression studies show that NRA1 regulates VEGF-induced angiogenesis and also the induction of integrins [75,76,77]. Interestingly, NR4A1 mediated induction of β4-integrin in HUVEC cells [2] and a subsequent study showed that induction of other integrins was also NR4A1-dependent [77].

### 3.8. Bone

NR4A1 is an important regulator of macrophage function, and Hamers and coworkers investigated the role of NR4A1 in bone marrow-derived macrophages (BMMs) in both wild-type and NR4A1-KO mice [78]. NR4A1 downregulates CXCL12 (SDF-1α) expression, enhances CX3CR1 expression, and represses NFkB, suggesting an anti-inflammatory effect of NR4A1 in BMMs. A subsequent report by this group [79] compared results of studies in wild-type classical NR4A-KO and novel NR4A1-KO (Cre) mouse models. It was demonstrated that the classical NR4A1-KO mice expressed the N-terminal aa1-117 region of NR4A1 whereas the NR4A1-KO (Cre) mice did not. Moreover, the classical NR4A1-KO mice (aa1-117) exhibited several functional and structural abnormalities not observed in wild-type or NR4A1-KO (Cre) mice [79]. In contrast, in both the classical and full-length NR4A1-KO mice there were decreased circulating Ly6C^low^ monocytes and decreased EGFR and VCAM1, which are bone marrow retention genes. Overall the report suggests that results obtained in NR4A1-knockout mice (whole body) expressing aa1-117 may need to be reinterpreted, particularly since this truncated form of NR4A1 (namely TR3β) is expressed and may be functional in some tissue/cell types, including human bone marrow cells and acute myeloid leukemia.

### 3.9. Autoimmunity

Loss of NR4A1 in T cells in mice is characterized by a highly proliferative phenotype and enhanced susceptibility to T cell-mediated inflammatory disease such as contact dermatitis and CNS autoimmunity [80]. NR4A1-deficient myeloid cells exhibit increased production of norepinephrine, and this resulted in acceleration of experimental autoimmune encephalomyelitis (AEA) and increased lumbar spinal cord Th expression [81]. Thus NR4A1 is an endogenous inhibitor in this model of induced multiple sclerosis and this is due, in part, to repression of norepinephrine and Th expression; a more recent study showed similar functions for NR4A1 in CNS autoimmunity [82]. It was also reported that NR4A1 was protective in a mouse model of lupus by inhibiting synaptic stripping by microglia [83].

### 3.10. Immune Responses

NR4A family members play a key role in regulating immune cell function and their expression is induced by T cell receptor signaling in the nucleus and NR4As play both overlapping and specific roles in T cells. NR4As enhance differentiation of CD4+ T cells into Treg cells in the thymus and peripheral tissues and this involves induction of Foxp3 and suppression of Th2 and Th17 cytokine genes [84,85,86,87,88]. NR4A1 (and NR4A3) acts to restrain B cell responses to antigen and interactions with T cells, and this is mediated by decreased expression of MYC and the basic leucine zipper ATF-like transcription factor (BATF) [89,90]. NR4A1 is also expressed in dendritic cells but loss of the receptor in mice did not significantly impact the development of dendritic cells in the spleen and lymph nodes; knockdown of NR4A1 in human monocyte-derived dendritic cells and murine dendritic cells enhanced inflammatory responses and T cell populations were increased [91]. Most studies with macrophages demonstrate that NR4A1 and other NR4As attenuate inflammation-induced responses [92]. Other examples of NR4A1-mediated effects in immune cells and resident tissues are discussed throughout this section.

### 3.11. Lung

In patients with pulmonary arterial hypertension (PAH), the pulmonary artery smooth muscle cells (PASMCs) exhibited increased proliferation and survival and decreased expression of NR4A1, NR4A2, and NR4A3 (protein and mRNA) compared to normal lung PASMCs [93]. In contrast, expression of the three NR4As in lung tissue is increased in patients with PAH compared to donors. Results of both human and mouse studies demonstrate that loss of NR4A1 in PASMCs results in decreased wound healing, increased cell proliferation, decreased axin, and increased β-catenin mRNAs. Studies in mice with hypoxia-induced pulmonary hypertension also demonstrate a protective role for NR4A1 [93].

### 3.12. Kidney

Several studies show that loss of NR4A1in the kidney enhances basal, genetic, and induced kidney damage. There was a decrease in NR4A1 in 20-month-old compared to 5-month-old mice and this correlated with increased renal tubular injury fibrosis, pSMAD2/3/SMAD2/3 ratio, collagen1, and Acta2 [27]. In addition, most of these same parameters were also increased in NR4A1-deficient mice, demonstrating the protective role of NR4A1 against unilateral ureteral obstruction (UUO)-induced fibrosis. Inhibition of the P13K/AKT pathway [94] and VEGF was associated with NR4A1-mediated inhibition of basal and unilateral ureteral obstruction-induced effects in mouse models [95]. In rat models loss of NR4A1 enhanced macrophage-mediated rat renal injury [96]. An in vivo study in mice and ITK-2 human kidney cells showed that UUO-induced effects in mice resulted in increased NR4A1, and fibrosis and similar results were observed in TGFβ-induced fibrosis in ITK-2 cells. Moreover, CsnB enhanced fibrosis [97]. This study did not examine the effects in NR4A1-deficient mice or cells and the results are in contrast to most other reports on NR4A1 and tissue injury.

### 3.13. Bladder

Urinary tract infection (UTI) of the bladder by UroPathogenic Escherichia Coli (UPEC) is a serious and common infection that is traditionally treated with antibacterials. It was recently observed in wild-type and NR4A1-KO mice that loss of NR4A1 increased the persistence of UPEC in bladder tissue, and this was accompanied by enhanced bacterial communities in the bladder of NR4A1-KO mice [98]. Thus NR4A1 plays a protective role in terms of decreasing UTI by acting on bacterial communities, and this response coupled with the effectiveness of treatment with CsnB is a highly novel approach for treating UTI.

### 3.14. Liver

The basal activity of NR4A1 in the liver is primarily derived from comparing liver function in normal and NR4A1-KO cells and in vivo. In FOX03-deficient mice and liver cells, loss of NR4A1 increased expression markers of hepatotoxicity [40]. Most studies show that NR4A1 plays a protective role in various stress-induced models of hepatotoxicity. For example, NR4A1 protects against homocysteine-induced hepatic steatosis, drug-induced liver injury, ischemic reperfusion injury, hypoxia–reperfusion injury, TGFβ-induced fibrosis, and genetic-induced liver damage [99,100,101,102,103,104,105,106,107].

### 3.15. Interactions with ROS

Reports on the potential NR4A1-mediated antioxidant activities in non-transformed tissues/cells are limited; however, there is some supporting data. For example, in vascular endothelial dysfunction the induction of ROS production is enhanced; this can be alleviated by overexpression of NR4A1 and the subsequent activation of genes that protect against ROS and nitric oxide production [108]. There is also evidence that NR4A1 “may be a sensor of oxidative stress and an inhibitor of vascular remodeling” [109], and a recent study showed that metformin inhibited myocardial ischemia–reperfusion injury and this is due, in part, to enhanced NR4A1 production and receptor-dependent induction of isocitrate dehydrogenase1 [110]. In a model of Parkinson’s disease, 1-methyl-4-phenylpyridinium (MMP^+^)-induced apoptosis and ROS in SH-SY5Y cells was inhibited by overexpression of YY1, which in turn induced NR4A1 expression [111]. These results suggest that NR4A1-mediated inhibition of ROS may be due to the activation of gene products that downregulate ROS or induce genes associated with enhance cellular reductant formation.

## 4. NR4A1 Ligands and Their Modulation of NR4A1-Dependent Responses

### 4.1. Introduction

Most but not all studies [112,113] confirm that NR4A1 plays a health-protective role in maintaining cellular homeostasis, as indicated above. Among the most commonly observedNR4A1/NR4A1 ligand-mediated anti-inflammatory responses is the inhibition of NFkB signaling, and this has been extensively reviewed [20,21,22,23,114,115]. Initial studies on the structures of NR4A proteins indicated that the ligand-binding domains contained bulky amino acid side chains and that these receptors were ligand-independent transcription factors [116]. This conclusion was consistent with the fact that endogenous ligands for orphan nuclear receptors such as NR4A have not been identified. However, a rapidly increasing number of structurally diverse compounds that directly interact with the ligand-binding domain of NR4A1 have been characterized [22,23,24] and act as agonists or inverse agonists that activate or inhibit expression of NR4A1-regulated genes, respectively. The agonist or inverse agonist activities of these ligands may be gene-, response-, and cell context-dependent, and this is not uncommon for ligands for other nuclear receptors. For example, tamoxifen, a selective estrogen receptor modulator (SERM), acts as a functional ER antagonist in breast cancer but exhibits functional ER agonist activities in the bone and endometrium [117,118]. This latter response has been associated with an increased incidence of endometrial cancer in women with breast cancer who were treated with tamoxifen for extended periods [119].

### 4.2. Effects of NR4A1 Ligands on Liver, Intestinal, Bladder, and Kidney Damage

Among the many structurally diverse compounds that bind NR4A1, the natural product cytosporone B (CsnB) has been the most widely used for determining effects on NR4A1-regulated non-cancer endpoints [120,121]. Thus the effect of this compound and other NR4A1 ligands serve as a model for the potential effects of naturally occurring dietary compounds that may act through NR4A1 to induce health benefits, and this will be discussed in the next section of this review. Table 1 summarizes several studies that have investigated the role of NR4A1 in multiple tissues/organs, and comparisons between wild-type NR4A and NR4A^−/−^ mice invariably show that the loss of NR4A1 results in enhanced tissue/organ damage. Moreover the use of CsnB or other NR4A1 ligands such as 6-mercaptorpurine (6-MP), bis-indole-derived compounds (CDIMs), celastrol, or the Gly-Pro-Ala (GPA) peptide further enhance the protective effects of NR4A1. Results of these studies in non-cancer tissues provide insights into cell or tissue damage that may also be observed as part of the aging process and which is protected by treatment with NR4A1 ligands.

In cultured human hepatic stellate cells expressing the PNPLA3 I148M variant, a risk factor for fibrogenic liver disease, there was increased TGFβ signaling and decreased NR4A1 expression compared to wild-type cells [107]. These cells exhibited multiple dysfunctional characteristics. A series of experiments in wild-type and NR4A1-KO mouse-derived cells showed that loss of NR4A1 enhanced induced fibrosis and steatosis and NR4A1 ligands such as CsnB protected against these responses in NR4A1-expressing liver cells/tissues [99,104,107]. The response patterns in mouse and in cell culture models of intestinal inflammation were similar to those observed in the liver, where induced inflammatory responses are enhanced in cells/tissues deficient in NR4A1 compared to wild-type, and NR4A1 and ligands such CsnB or 6-MP ameliorated cell damage phenotypes in wild-type NR4A1-expressing intestines [72,73,74,122]. A novel role for NR4A1 was observed in urinary tract infection (UTI) by uropathogenic *E. coli* (UPEC), where loss of NR4A1 in mice resulted in enhanced bacterial infection in the bladder [98] and CsnB decreased infection in wild-type mice and in cell culture, and this represents a novel mechanistic and therapeutic approach for treating UTI. Unilateral ureteral obstruction (UUO) in mice induced NR4A1 and enhanced fibrosis and interstitial kidney damage and CsnB enhanced this response, and similar enhancement was observed in a TGFβ-induced response in vitro [97]. The pro-fibrotic activity of CsnB contrasts with most other effects observed for this NR4A1 ligand. Moreover, another publication on UUO-induced renal damage reported that both NR4A1 and CsnB exhibited protective effects [94], though reasons for the differences between these studies are unclear. Moreover, CsnB also inhibited induced kidney fibrosis in a mouse model [104].

### 4.3. Effects of NR4A1 Ligands on Pulmonary Bone and CNS Damage

Pulmonary arterial hypertension (PAH) is a serious disease in humans and NR4A1 is downregulated in PASMCs in humans with PAH, and overexpression of NR4A1 in PASMCs blocked cell proliferation and migration [93]. LPS-induced pulmonary damage and activation of inflammatory factors in rats was inhibited by pretreatment with pterostilbene, which was shown to interact with NR4A1 [123]. Mouse model studies demonstrated that hypoxia-induced PAH-like symptoms were mitigated by treatment with CsnB. Influenza-induced adverse effects on pulmonary function are also ameliorated after treatment with CsnB, which acts on both lung tissue and macrophages, and this activity is associated with the induction of type I interferon [124]. Another study showed that 6-MP also inhibited progression of pulmonary hypertension, where NR4A1 induced bone morphogenic protein LCBMP signaling and decreased inflammation and proliferation in microvascular endothelial cells [125]. Inflammation associated with osteoarthritis in humans was accompanied by increased expression of NR4A1 and overexpression of several MMPs, COX-2, and iNOS [126]. In a rat model of osteoarthritis, CsnB inhibited the same set of inflammatory genes and protected against inflammation.

The role of NR4A2 and effects of NR4A2 ligands on different aspects of neuronal toxicities including learning and memory and Parkinson’s disease have been extensively investigated [52,127,128,129,130] whereas this is not the case for NR4A1 ligands. MPTP induced neurotoxic effects in a mouse model of Parkinson’s disease and is accompanied by induction of NFkB and progressive loss of dopaminergic neurons in the substantia nigra in mice [129]. Protection from the induced neurotoxic responses was observed after treatment with the NR4A1 ligand (1,1-bis(3′-indolyl)-1-(4-hydroxyphenyl)methane (DIM-4-OH) and the corresponding 4-chlorophenyl analog (DIM-4-CI), which is an NR4A2 ligand [130]. PC12 rat adrenal pheochromocytoma cells were used as a model of Parkinson’s disease for investigating the role of NR4A1 in modulating the effects of MPP^+^-induced inflammation (and oxidative stress) and induction of TNFα, MCP-1, IL-6, and NFkB. The loss of NR4A1 mimicked the effects of MPP^+^, and CsnB inhibited MPP^+^-induced inflammatory gene product formation and also decreased MPP^+^-induced oxidative stress in wild-type cells [39]. NR4A1 and CsnB also ameliorate induced inflammation in microglia and specifically protect dopaminergic neurons. A key mechanistic aspect associated with the NR4A1/CsnB-mediated response was due to inhibition of IKBα-phosphorylation [131]. In vivo studies in wild-type and NR4A1-KO mice showed that NR4A1 suppressed cytokine expression and NO production in microglia [132]. Moreover, the loss of NR4A1 enhanced experimental autoimmune encephalomyelitis in mice compared to wild-type mice, and in mice expressing NR4A1, treatment with CsnB inhibited the course of this disease model for multiple sclerosis. Carpenter and colleagues reported that in a mouse model, cocaine-activated expression of NR4A1 and several downstream genes including cocaine- and amphetamine-regulated transcript peptide (Cartpt) and mediated cocaine-induced behavior [133]. Moreover, CsnB activation of NR4A1 suppressed cocaine behavior, and it was suggested that NR4A1 ligands such as CsnB may be of therapeutic value for treating cocaine addiction.

**Table 1 nutrients-17-02709-t001:** Effects of NR4A1 ligands on NR4A1-regulated responses and genes.

Ligand [Ref]	Tissue/Cell Type	Response/Genes
CsnB [107]	Liver	PNLPLA3 I148M variant in human hepatic stellate cells increased dysfunction and decreased NR4A1 but this was increased by CsnB
CsnB [104]	Liver	NR4A1 inhibited TGFβ-induced fibrosis and CsnB enhanced inhibition (also skin, lung, kidney)
CsnB [99]	Liver	Homocysteine induced hepatic steatosis, which is blocked by CsnB
CsnB/6-MP [72]	Intestine	Induced smooth muscle cell phenotype enhanced in NR4A^−/−^ mice; decreased by CsnB and 6-MP
CsnB/6-MP [73]	Intestine	TGFβ-induced fibrogenesis in myofibroblasts enhanced with loss of NR4A1: CsnB and 6-MP decreased response in wild-type cells
Csn [75]	Intestine	DSS-induced colitis inhibited by CsnB NR4A-TLR-1R via NR4A-TRAF6 interactions
GPApeptide [122]	Intestine	GPA inhibited NFkB activation and intestinal inflammation in DSS-induced colitis
CsnB [98]	Bladder	NR4A1 protects against urinary tract infections and CsnB inhibits bacterial infection
CsnB [97]	Kidney	Induced unilateral ureteral obstruction in mouse UUO-induced kidney fibrosis and NR4A1 and CsnB enhanced the response
CsnB [94]	Kidney	NR4A1 is protective against UUO-induced fibrosis and CsnB is also protective
Pterostilbene [123]	Lung	Pterostilbene inhibited LPS-induced inflammation and damage in the rat lung in vivo
CsnB [93]	Lung	NR4A1 downregulated in pulmonary arterial hypertension (PAH), CsnB inhibits symptoms in hypoxia-induced mouse model
CsnB [124]	Lung	CsnB inhibits influenza-induced pulmonary damage and is associated with induction of type I interferon
6-MP [125]	Lung	6-MP inhibited progression of pulmonary hypertension and was associated with NR4A1 activation of bone morphogenic protein
CsnB [126]	Bone	NR4A1 elevated in osteoarthritis and CsnB inhibited IL-1β-induced inflammatory genes and decreased osteoarthritis in a rat model
CsnB [39]	CNS	CsnB inhibited MPP^+^-mediated inflammatory and oxidative stress genes in rat PC12 pheochromocytoma cells
CsnB [131]	CNS	CsnB inhibits inflammation/inflammatory genes and reverses MPTP-induced TH positive neurons and Iba-1 positive neurons
CsnB [132]	CNS (autoimmune)	NR4A1 suppressed cytokine production and NO in mouse microglial cells and NR4A1 alone and in combination inhibited the progress of AEA-induced disease in a model of multiple sclerosis
CsnB [133]	CNS	Cocaine-induced NR4A1 and cartpl gene expression NR4A1/CsnB modify cocaine-induced behavior
CsnB [134]	Endometriotic	Csn inhibits TGFβ-induced fibrosis
DIM-4-OH and DIM-3-CI-4-OH-5-OCH_3_ [135]	Endometriotic	NR4A1 is pro-endometriotic and DIM compound inhibits mTOR, proliferation, and fibrosis
CsnB [136]	Cardiovascular	Loss of NR4A1 in hypocholesterolemia enhances IL-6 and MCP-1 and CsnB inhibits cholesterol-induced IL-6 and MCP-1
CsnB [137]	Cardiovascular	Hypocholesterolemia induced platelet activation and thrombus inhibited by NR4A1 and CsnB further protects—due, in part, to cAMP phosphorylation of VASP
CsnB [138]	Cardiovascular	NR4A1 decreases atherosclerotic responses and CsnB enhances their effects in mouse models of hypocholesterolemia
6-MP [139]	Cardiovascular	NR4A1/6-MP protect from restenosis-induced neointima formation and inhibition of cell proliferation
Celastrol [140]	Cardiovascular	Development of carotid plaque in wild-type ApoE^−/−^/NR4A1-KO mice inhibited by NR4A1 and enhanced by celastrol by inhibiting bcat in macrophages
CsnB [141]	Cardiovascular	Cardiac allograft rejection in mice was decreased by NR4A1 and enhanced by CsnB by targeting by infiltrating CD4+ T cells and inducing Treg cell differentiation
CsnB [142]	Inflammation	CsnB differentially inhibited LPS-induced NFkB in human macrophages
CsnB [143]	Inflammation	CsnB inhibited NFkB-mediated inflammation in a mouse model of sepsis
CsnB [144]	Inflammation	Ly6C^high^ monocytes contribute to a mouse model of arthritis and in the presence of Ly6C^low^ monocytes Csn inhibits progression of arthritis
AEA [145] DIM-4-CI [145] CsnB [145]	Inflammation	AEA is a dual NR4A1/2 ligand that inhibits IL-1β-induced cytokines (e.g., CCL_2_) in vascular smooth muscle cells, as do other ligands
TokinolideB [146]	Inflammation	Tokinolide induces NR4A1 nuclear export and inhibits inflammation in a mouse model of hepatitis through mitochondria autophagy pathways
Celastrol [147]	Inflammation	Induces nuclear export of NR4A1, which interacts with TRAF2 to inhibit inflammatory signaling
6-MP [91]	Inflammation	Decreased dendritic cell activation, decreased production of interferonƔ
CsnB [120]	Metabolic disease /Liver	CsnB elevated blood glucose levels and activated hepatic gluconeogenesis
TMPA [148]	Metabolic disease /Liver	TMPA decreased blood glucose levels and reversed insulin resistance
DIM-3,5 analogs [149]	Metabolic disease (muscle)	DIM-3,5 enhanced expression of GLUT4 and glycolytic genes and increased glucose uptake in muscle cells
CsnB [150]	Eye	Inhibited subretinal fibrosis and macrophage-to-myoblast transition

### 4.4. Effects of NR4A1 Ligands on Endometriosis and Cardiovascular Damage

Endometriosis is also a highly inflammatory disease characterized by cell growth, migration, and fibrosis. Inhibition of NR4A1 expression in stromal cells increased TGFβ-induced fibrosis and in NR4A1-expressing cells CsnB inhibited fibrosis [134]. Similar results for NR4A1 were observed in epithelial and stromal endometriotic cells, where NR4A1 knockdown resulted in activation of mTOR signaling and induction of α-smooth muscle actin and related fibrotic genes. Moreover, treatment of these cells with bis-indole-derived NR4A1 ligands 1,1-bis(3′-indolyl)-1-(4-hydroxyphenyl)methane (DIM-4-OH) and the 3-chloro-4-hydroxy-5-methoxyphenyl analog (DIM-4-OH-3-CI-4-OCH_3_) inhibited cell proliferation, mTOR signaling, and fibrotic gene expression [135]. The results suggest that NR4A1 is a pro-endometriotic factor in endometriosis and NR4A1 ligands act as inverse agonists, and similar results have been observed in solid tumor-derived cells in culture and in vivo [14].

NR4A1 ligands play an important protective role in cardiovascular disease [95]. Higher expression of NR4A1 is observed in leukocytes from patients with hypocholesterolemia and in cell culture cholesterol induces NR4A1 and loss of the receptor enhances expression of inflammatory genes such as IL-6 and MCP-1 [136]. CsnB treatment inhibits expression of cholesterol-induced IL-6 and MCP-1. Using human NR4A1 expression data and platelets from patients with hypocholesterolemia and mouse models, it was demonstrated that loss of NR4A1 (in mice) enhanced thrombus formation, microvascular microthrombi obstruction, and platelet activation. In cells/tissue expressing NR4A1, CsnB inhibited activation of hypocholesterolemic platelets. The mechanisms of NR4A1/CsnB protective effects involved enhancement of cAMP levels and subsequent phosphorylation of vasodilator-stimulated phosphoprotein (VASP) [137]. NR4A1 also decreased atherosclerotic plaque formation in ApoE^−/−^ mice maintained on a high-fat/high-cholesterol diet, and CsnB further enhanced the protective effects of NR4A1 in this model [138]. NR4A1 protects from restenosis, which can accompany coronary interventions and is associated with proliferation of smooth muscle cells and neointima formation [139] 6-MP also further inhibits neointima formation and this is accompanied by inhibition of PCNA, a marker of cell proliferation and induction of the cell cycle inhibitor p27^Kip1^ [139]. Using ApoE^−/−^ and NR4A-KO mice, another study showed that carotid plaque formation was enhanced in NR4A1-deficient mice, and this was associated with enhanced macrophage-mediated inflammatory and oxidative responses [140]. These responses were ameliorated by NR4A1-dependent inhibition of Bcat1 expression. The NR4A1 ligand celastrol also stabilized formation of atherosclerotic plaques in mice. CsnB was also investigated for a possible role in modulating acute cardiac allograft rejection in a mouse model [141]. NR4A1 was primarily expressed in intragraft-infiltrating CD4+ T cells and associated with apoptosis, differentiation, and T cell dysfunction. In a mouse model, CsnB decreased the allograft rejection response by targeting Treg cells and CD4+ T cells.

### 4.5. Effects of NR4A1 Ligands on Inflammation, Metabolic Diseases, and the Eye

Induced inflammatory responses are intimately linked to many NR4A-regulated responses in many tissues, as outlined in Table 1, and a few additional examples include the following: NR4A1 was more highly expressed in human macrophage cells cultured with macrophage colony-stimulating factor (M-MDM) compared to culture with granulocyte/macrophage-colony stimulating factor (GM-MDM). CsnB suppressed LPS-induced inflammatory responses (e.g., TNF and IL-6) in GM-MDMs and this was associated with decreased NFkB nuclear uptake; however, increased levels of IL-10 were also observed [142]. In addition, CsnB also inhibited sepsis in a mouse model, and this was associated with inhibition of NFkB and downstream inflammatory genes [143]. Results from NR4A1-KO mice indicate that inflammatory Ly6C^high^ monocytes contribute to a mouse model of arthritis; in NR4A wild-type mice, CsnB increases levels of CD4+, CD25+, and FOXP3+ Treg cells in the presence of Ly6C^low^ monocytes to inhibit progression of arthritis [144]. A recent report showed that the endocannabinoid anandamide AEA interacts with NR4A1 and NR4A2 and is a dual receptor ligand. AEA inhibited IL-1β-induced cytokines such as CCL2 in vascular smooth muscle cells [145]. Similar results were observed for CsnB and DIM-4-CI, which have been characterized as NR4A1 and NR4A2 ligands, respectively. TokinolideB is a phthalide isolated from *Angelica sinensis* that binds NR4A1 [146]; like celastrol [147], tokinolide induces nuclear export of NR4A1 and exhibits anti-inflammatory activity in a mouse model of hepatitis. The effects involve mitochondrial interactions with TRAFα to induce autophagy and this results in inhibition of LPS-induced inflammation. 6-MP decreased dendritic cell activation and inhibited interferonƔ production by allogeneic T cells [91].

Results of NR4A1 knockdown or overexpression demonstrate that this receptor plays a key role in protection from metabolic diseases; studies on the effects of NR4A1 ligands and their influence on diabetes and related responses are limited. CsnB induces blood glucose levels and activates hepatic gluconeogenes [120] whereas another CsnB analog, ethy l2-[2,3,4-trimethoxy-6-(1-octanoyl)phenyl] acetate (TMPA), decreased blood glucose levels and reversed insulin resistance [147]. Thus two structurally related compounds (CsnB/TMPA) exhibit inverse activities on metabolic disease, and it is possible that the prodiabetic activity of CsnB may be due to other modes of action, such as enhancement of NR4A1 nuclear export. Another study reported that DIM-3,5 analogs were antidiabetic and induced GLUT4, glucose uptake, and glycolytic genes in C2C12 muscle cells [148]. A recent study reported another application of CsnB where subretinal fibrosis was inhibited by NR4A1 [149]. Knockdown of NR4A1 promoted macrophage-to-myoblast transition (MMT) whereas CsnB inhibited this pathway and protected from subretinal fibrosis [150]. These results show that for most responses NR4A1 ligands enhance the health-promoting effects of NR4A1 in cell culture and animal models, as illustrated in Figure 4.

## 5. NR4A1 as Nutrient Sensor for Health-Promoting Polyphenolics

### 5.1. Introduction

It has previously been hypothesized that NR4A1 is protective against aging [151] and the age-dependent expression of NR4A1, and its interactions with key genes involved in aging support this hypothesis. There is also extensive evidence showing the protective effects of NR4A1 expression against multiple responses associated with premature aging, including inhibition of inflammation, enhanced immune cell functions, and decreased autoimmunity and damaged cells from multiple organs and tissues. Moreover, NR4A1 ligands such as CsnB enhance the health-promoting responses and genes regulated by NR4A1. These observations suggest that since NR4A1 is a receptor it is possible that one of its functions could be as a receptor for health-promoting dietary compounds such as the polyphenolics in fruits, nuts, and vegetables. These compounds include flavonoids, resveratrol, and other polyphenolics that are known to be associated with decreased mortality and lower incidence of cardiovascular, metabolic, and neurotoxicity diseases such as dementia and Alzheimer’s and Parkinson’s disease [5,6,7,8,152,153,154].

### 5.2. Flavonoids and Resveratrol Bind NR4A1 and Inhibit Endometriosis

The linkage between polyphenolics and NR4A1 and their role in promoting health was first demonstrated in a study showing that the flavonoids quercetin and kaempferol bind NR4A1 and exhibit NR4A1-dependent transactivation [155]. Moreover, a direct comparison of the flavonoids with CDIM NR4A1 ligands showed that they exhibited comparable activity as inhibitors of endometriotic cell growth in both in vitro and in vivo models [135]. For example in endometriotic cells, the CDIM compounds and the flavonoids (quercetin/kaempferol) decreased NR4A1-dependent transactivation (using a GAL4-NR4A1 construct), inhibited endometriotic cell proliferation, induced apoptosis, and decreased mTOR signaling and fibrotic genes, including α-smooth muscle actin (αSMA), fibronectin (FN), connective tissue growth factor (CTGF), and collagen type 1α (COL1A1) [155]. These responses were also observed after knockdown of NR4A1 by RNA interference, demonstrating that like the CDIM compounds, quercetin and kaempferol acted as inverse NR4A1 agonists in endometriotic cell lines. Similar results were observed for these compounds in Rh30 rhabdomyosarcoma cells, where the flavonoids inhibited growth and induced apoptosis [156]. Subsequent studies showed that in addition to quercetin and kaempferol, 18 hydroxylated flavones also bound NR4A1, and this included several individual compounds found in fruits, nuts, and vegetables, including galangin, chrysin, luteolin, apigenin, and baicalein. K_D_ binding values were variable but most of the compounds noted above exhibited K_D_’s < 10 µM, with the lowest values (0.36 µM) observed for galangin [157].

### 5.3. Quercetin Effects Mimic Those of Synthetic NR4A1 Ligands

The health-promoting effects of flavonoids have been extensively investigated and the number of citations (PubMed) for quercetin alone is >31,000. A comparison shows that the effects observed for quercetin mimic those reported for other NR4A1 ligands, suggesting a role for NR4A1 in mediating quercetin-induced responses (Figure 5). For example, numerous studies show that quercetin inhibits induced hepatic fibrosis and hepatic stellate cell activation [158,159,160,161,162] by modulating many of the same responses observed for NR4A1 ligands, and the effects of quercetin on kidney fibrosis [163,164,165] were also similar to those observed for NR4A1/ligands (Table 1). Quercetin also inhibits cyclophosphamide-induced urotoxicity and expression of inflammatory genes [166], and several studies reported antibacterial effects of quercetin [167]. Quercetin also inhibits induced neurotoxicity and its associated inflammation [168,169] and also lung damage by inhibiting fibrosis, oxidative stress, and inflammation and there is also evidence of a role for NRF2 [170,171,172,173]. Quercetin inhibits mouse models of intestinal inflammation [174]; for example, dietary quercetin inhibited DSS-induced intestinal damage by multiple pathways, including improve colonic permeability and inhibition of IL-6, TGFβ, and other inflammatory markers [175]. There are extensive studies on the protective effects of quercetin on most types of cardiovascular damage and diseases [176,177,178]. For example, quercetin inhibited LPS-induced sepsis through inhibition of multiple pro-inflammatory cytokines and NFkB [179]. Like NR4A1 and its ligands, quercetin also exhibited antidiabetic activity [180]. Quercetin induced GLUT4 and uptake of glucose in C2CI2 muscle cells and inhibited gluconeogenesis in liver cancer cells [181] and inhibited oxidative damage and potentiated insulin secretion in pancreatic β cells [182]. Quercetin inhibited inflammatory markers, enhanced M2 polarization in macrophages [183,184], and inhibited LPS-induced activation of dendritic cells, which was accompanied by inhibition of pro-inflammatory cytokines and chemokines [185]. These results (Figure 5) demonstrate that quercetin, an NR4A1 ligand, exhibits anti-inflammatory activities similar to those observed for CsnB (Table 1) and other synthetic NR4A1 ligands. Moreover, the same holds true for other flavonoids that bind NR4A1. A recent study showed that the flavonoid baicalein activated NR4A1 to inhibit ovalbumin-induced allergic rhinitis in mice [186] and it is likely that this response was also due, in part, to baicalein acting as an NR4A1 ligand [153]. It is also likely that NR4A1 plays a role in mediating some of the health-protective responses associated with quercetin, kaempferol, baicalein, and related compounds that bind NR4A1 [153].

### 5.4. Resveratrol Exhibits NR4A1 Ligand-like Activities

Many other health-promoting phenolics and their interactions with NR4A1 have not been investigated; however, a recent study showed that resveratrol, a polyphenolic found in grapes, red wine, peanuts, and chocolate also bound NR4A1 and inhibited NR4A1-mediated pro-oncogenic pathways in lung cancer cells [187]. Resveratrol, like quercetin and other polyphenolics, has been linked to multiple health benefits in humans [188]. Resveratrol induces many of the same responses observed for NR4A1 and its ligands in cell culture and animal models (Table 1) and is also associated with anti-aging effects [189,190]. For example, resveratrol inhibited intestinal fibrosis and intestinal barrier dysfunction [191,192], inhibited liver and kidney fibrosis [193,194,195,196], exhibited therapeutic value for treating diabetes [197,198], inhibited cardiovascular damage [199,200], protected against neurotoxicity [201,202], inhibited urinary tract infections and bladder damage [203,204,205,206], reversed lung damage [207,208], and modulated immune cell functions [209,210,211]. Moreover, resveratrol, quercetin, and other flavonoids have been extensively used in clinical trials to treat some of these health problems. Studies in humans demonstrate an association between consumption of polyphenolics and some protection against aging-related diseases in humans and this has also been demonstrated in cell culture and animal models [5,6,7,8,212,213,214]. NR4A1 and its ligands have been extensively characterized for their protective effects against stressors/inflammation-induced tissue damage. Based on the similarities observed for the protective effects of NR4A1 ligands (Table 1) and the newly characterized polyphenolic NR4A1 ligands (flavonoids and resveratrol), it is possible that NR4A1 may also act as a nutrient sensor and mediate some of effects of polyphenolics as “geroprotectors” [214].

## 6. Conclusions

This review demonstrates that the orphan nuclear receptor NR4A1 protects against aging and interacts with other gene products involved in the aging process. Results of NR4A1-KO mouse and cell culture studies show that NR4A1 protects against tissue damage, including inflammation, fibrosis, and barrier dysfunction, and this is consistent with initial studies on NR4A1 and other members of this subfamily as genes that are induced in response to stressors and inflammatory agents to ameliorate their effects [18,19]. The functional effects of synthetic and diet-derived polyphenolic NR4A1 ligands are similar; however their genomic pathways show some differences, which are not surprising for compounds that act through nuclear receptors. Nuclear receptor ligands exhibit differences due to several factors, including their unique structure-dependent interactions with receptors, which is accompanied by conformational differences in ligand–receptor structures. This suggests that NR4A1 ligands, synthetic or natural act, as selective NR4A1 modulators and this is consistent with some of the differences in their tissue-specific activities. Thus the synthetic NR4A1 ligands enhance the protective effects of NR4A1 and the discovery that flavonoids and resveratrol are also NR4A1 ligands suggests that the anti-aging and health benefits of these polyphenolics may be due, in part, to their activities as NR4A1 ligands. The linkage between polyphenolics and NR4A1 as a nutrient sensor for this class of “anti-aging” compounds is based primarily on a few studies. Further confirmation is required to delineate the contributions of NR4A1 in mediating the health-protective effects of dietary phenolics and other compounds and for the development of dietary supplements that will enhance these effects.

## Figures and Tables

**Figure 1 nutrients-17-02709-f001:**
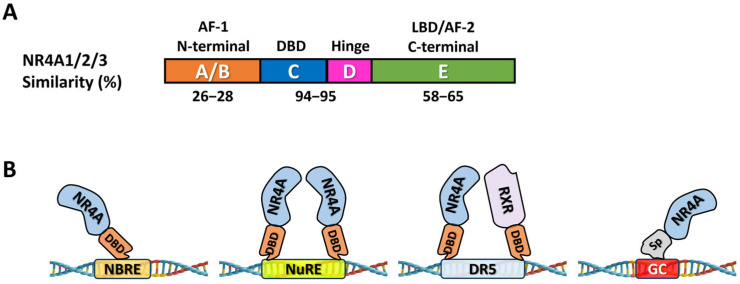
NR4A subfamily of NRs. (**A**) Domain structure of NR4A shows the percentage (%) of similarity between NR4A1, NR4A2, and NR4A3 in 3 domains. (**B**) Protection interactions of NR4A1 and other ligands with DNA response elements.

**Figure 2 nutrients-17-02709-f002:**
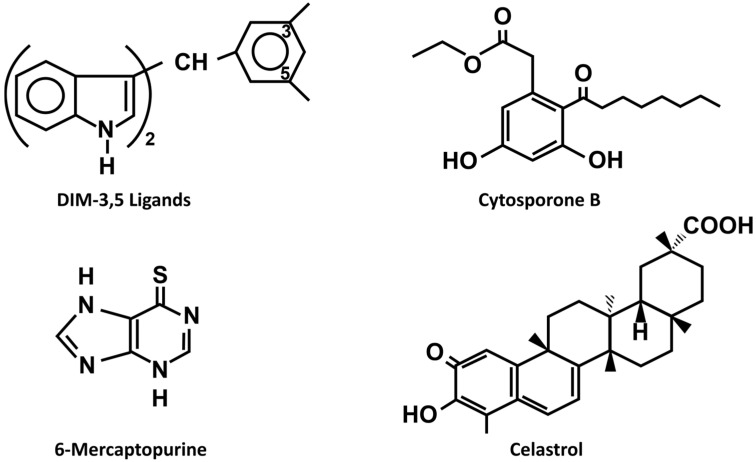
Examples of synthetic and natural product-derived chemicals that bind NR4A1 [22,23,24].

**Figure 3 nutrients-17-02709-f003:**
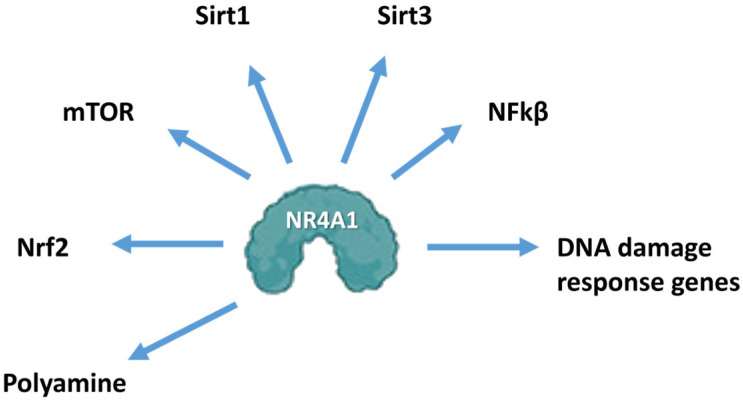
Interactions of NR4A1 with various genes/gene products associated with aging.

**Figure 4 nutrients-17-02709-f004:**
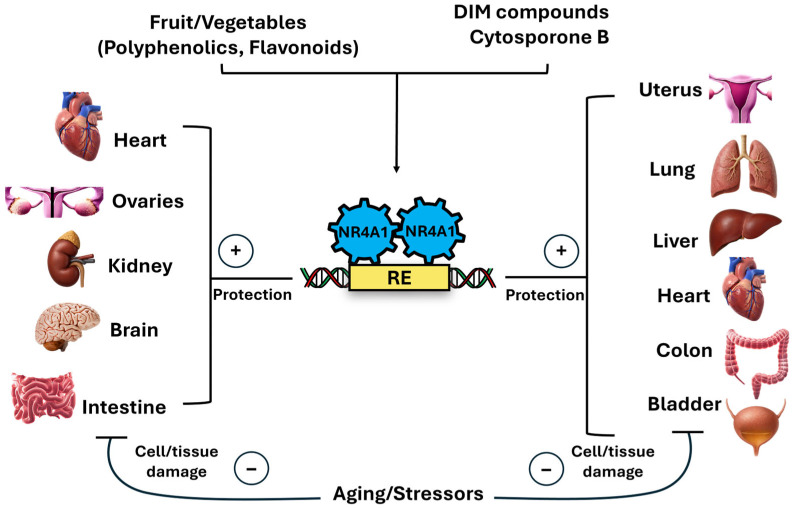
Proposed model for NR4A1 as a nutrient sensor that plays a constitutive and ligand-induced role in preventing tissue damage. NR4A1 may also contribute to the overall health benefits of dietary polyphenolics.

**Figure 5 nutrients-17-02709-f005:**
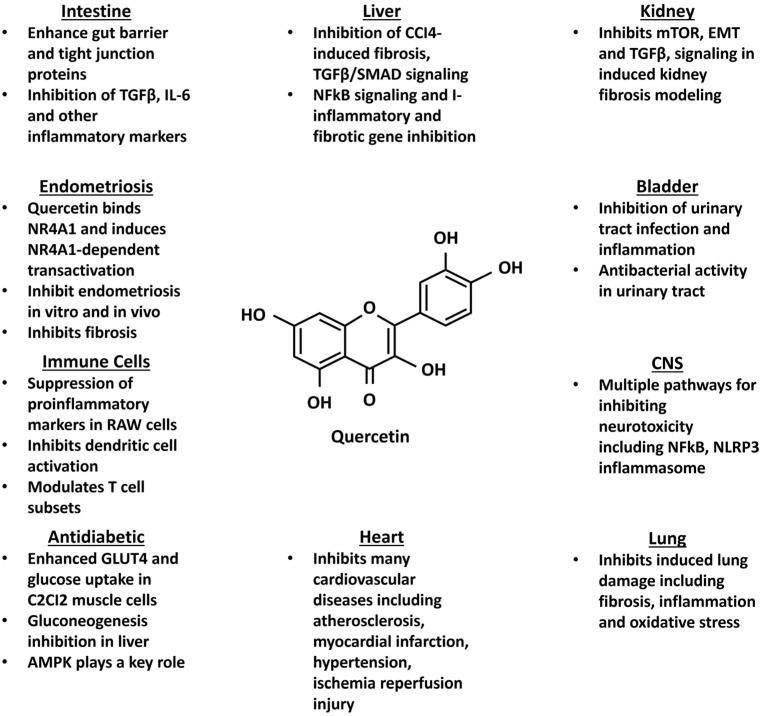
Health-promoting effects of quercetin in various cell culture and laboratory animal models and their similarity to other synthetic NR4A1 ligands (Table 1).

## Data Availability

No new data were created or analyzed in this study. Data sharing is not applicable to this article. Material used in this review was obtained primarily from PubMed searches.

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
