# Peer review of "NR4A1 Acts as a Nutrient Sensor That Inhibits the Effects of Aging"

_nutrients, 2025, doi:10.3390/nu17162709_

Round 1

Reviewer 1 Report

Comments and Suggestions for Authors

The paper relates a highly relevant topic, as Orphan nuclear receptor 4A1 being expressed in response to stressors including inflammatory agents, plays a key role in ifluencing the rate of aging in both humans and animals. My questions/remarks are as follows:

  • Why has not the comparison extended to the assessment of the antioxidant status, bioavailability, bioutilization apart from inflammatory diseases, immune dysfunction and fibrosis?
  • How could the adverse effects of NR4A1-deficiency be mitigated? What kind of treatments might be of beneficial impact the counteract this factor?
  • Please provode more in-depth explanation for the findings underscoring the key factors and evidences of the NR4A1-dependent inhibition of the aging process.
  • It is mentioned in the paper that recent studies show that flavonoids such as quercetin and kaempferol and resveratrol bind NR4A1 and exhibit protective NR4A1-dependent inhibition of endometriosis and cancer. There are several scientific evidences certifying the contrary of this observation, please mention more findings on this issue, as well.
  • How could the major conclusion of the paper incorporated in the practice of „healthy food industry”? How might NR4A1 as a potential dietary sensor be more efficintly 

Author Response

Reviewer 1

  • Why has not the comparison extended to the assessment of the antioxidant status, bioavailability, bioutilization apart from inflammatory diseases, immune dysfunction and fibrosis?

Response: The major focus of this review is to demonstrate that NR4A1 and its ligands play a role in mitigating effect of cellular damage associated with aging in cell culture and animal models. There is limited data in non-cancer cells on the antioxidant effects of NR4A1/ligands. However, there is some evidence that NR4A1 inhibits oxidative stress and this is now incorporated as a separate section into the revised manuscript (lines 330-342).

  • How could the adverse effects of NR4A1-deficiency be mitigated? What kind of treatments might be of beneficial impact the counteract this factor?

Response: NR4A1 deficiency can be overcome in some cell types by inducing expression of this gene however, this has not been well described.

  • Please provode more in-depth explanation for the findings underscoring the key factors and evidences of the NR4A1-dependent inhibition of the aging process.

Response: A brief summary of the evidence supporting a role for NR4A1 as an anti-aging receptor has now been included in the overall summary of the Review and this is included in lines 612-620 and 638-642.

  • It is mentioned in the paper that recent studies show that flavonoids such as quercetin and kaempferol and resveratrol bind NR4A1 and exhibit protective NR4A1-dependent inhibition of endometriosis and cancer. There are several scientific evidences certifying the contrary of this observation, please mention more findings on this issue, as well.

Response: The focus on the quercetin/kaempferol effect on endometriosis and the resveratrol paper is to illustrate the linkage of these health-protecting polyphenolic to NR4A1. This linkage coupled with the overlapping effect of polyphenolics such as quercetin (Fig. 5) with bona fide NR4A1 ligands suggests that NR4A1 to may be a sensor of polyphenolics. It is true that this data is limited and will require extensive confirmatory studies and this is now indicated in the revised manuscript (lines 638-642) .

  • It How could the major conclusion of the paper incorporated in the practice of „healthy food industry”? How might NR4A1 as a potential dietary sensor be more efficiently 

Response: Practical applications for the health food industry may be the availability of specific foods and/or nutriceuticals enriched in NR4A1 ligands such as polyphenolics. This is now noted in the revised manuscript (lines 612-620 and 638-642).

Reviewer 2 Report

Comments and Suggestions for Authors

This review provides very interesting information regarding the role of NR4A1 in relation to aging. However, the following points should be addressed before publication.

(1)    The aim of this review should be explained in “Abstract” and “Introduction”.
(2)    The originality and novelty of this review are also necessary.
(3)    How about “Limitations” of this review?
(4)    How about “Further directions”?
(5)    Figure 1 does not correspond to the explanation (Line 28-53) in Introduction.
(6)    All the references should be carefully re-checked to collect several misquotations. 
(7)    The number of author’s self-citation (Safe S. et al.) appears to be too many (19 papers).  

Author Response

Reviewer 2

(1)    The aim of this review should be explained in “Abstract” and “Introduction”. Response: The aim of the review is now indicated in the Abstract and Introduction.

(2)    The originality and novelty of this review are also necessary.
(3)    How about “Limitations” of this review?

Response: It was previously hypothesized that NR4A1 may protect from aging and this review provides detailed evidence supporting this hypothesis and includes new data linking the effects of polyphenolics to their activities as NR4A1 ligand. While these new studies support a role for this receptor as an anti-aging factor this will have to be further investigated due to the limited amount of data. These issues are emphasized in the review (lines 612-620 and 638-642).

(4)    How about “Further directions”?

Response: As indicated support for the role of NR4A1 in aging and as a nutrient sensor will require additional studies.

(5)    Figure 1 does not correspond to the explanation (Line 28-53) in Introduction.
Response: This has now been corrected.

(6)    All the references should be carefully re-checked to collect several misquotations. 
(7)    The number of author’s self-citation (Safe S. et al.) appears to be too many (19 papers).

Response: We have reviewed the reference to correct the misquotations and have reduced the self-citation from 19 to 14.

Reviewer 3 Report

Comments and Suggestions for Authors

Dear Authors,

Thank you for submitting your manuscript entitled “NR4A1 Acts as a Nutrient Sensor that Inhibits the Effects of Aging.” The topic is both timely and relevant, and your review provides a well-referenced overview of NR4A1 and its potential roles in modulating aging-related processes. However, several important issues need to be addressed to improve the scientific clarity and rigor of the manuscript.

Comments:

  • The title is engaging but does not fully reflect the review-based nature of the manuscript.
  • The manuscript lacks a description of the methodology used for literature selection and review.
  • The cited studies are presented descriptively without deeper critical analysis or discussion of conflicting data.
  • The conclusions go beyond the available evidence—claiming a unique role of NR4A1 as a nutrient sensor that slows aging is a bold interpretation based on limited and mostly indirect findings.
  • The evidence presented relies largely on animal and cell culture models, with indirect outcomes (e.g., markers of inflammation or fibrosis); direct data on NR4A1 ligands influencing lifespan or aging rates are missing.
  • There is a typographical error: “Pulomary” should be corrected to “Pulmonary.”
  • The presented data from mouse and cell models do not directly prove that NR4A1 protects against aging.
  • The assumption that the health-promoting effects of flavonoids are due to NR4A1 activity is speculative and does not account for the many other known biological targets of these compounds.

Author Response

Reviewer 3

  • The title is engaging but does not fully reflect the review-based nature of the manuscript.
  • The manuscript lacks a description of the methodology used for literature selection and review.

Response: I chose the title based on the linkage between the comparable effects of bona fide NR4A1 ligands vs nutrient-derived NR4A1 ligands and am confident that this relationship will be strengthened in the future. PubMed was the prime source for assembling the content and references for this review and this is now indicated in the
“Data availability statement”.

  • The cited studies are presented descriptively without deeper critical analysis or discussion of conflicting data.

Response: A brief descriptor of the pathway/mechanism involved in the various NR4A1/NR4A2 ligand responses was included in the Tables. More detailed discussion would be beyond the scope of this review. In terms of the tissue-specific protective effects of NR4A1 and its ligands the data were highly consistent. Some conflicting results were included (e.g., references # 79, 94 vs 97).

  • The conclusions go beyond the available evidence—claiming a unique role of NR4A1 as a nutrient sensor that slows aging is a bold interpretation based on limited and mostly indirect findings.

Response: I agree that the evidence for NR4A1 as a nutrient sensor is limited and have added statements that point out this limitation. The review initially highlighted only two studies demonstrating that two flavonoids (quercetin and kaempferol) and resveratrol bind NR4A1 and act as NR4A1 ligands. However, over 20 flavonoids bind NR4A1 and are likely to exhibit NR4A1 activities similar to that described for kaempferol and quercetin and a recent confirmatory study (2025, ref 186) on baicalein has been added.

  • The evidence presented relies largely on animal and cell culture models, with indirect outcomes (e.g., markers of inflammation or fibrosis); direct data on NR4A1 ligands influencing lifespan or aging rates are missing.

Response: The role of NR4A1 and aging were primarily derived from studies in mice with some supporting data from humans. Although effects of synthetic or modified NR4A1 ligands and aging have not been reported there are some studies on the correlation of NR4A1-active flavonoids with enhanced health in humans and multiple studies in animal models and cell culture. Moreover, among those nutrients that are most frequently correlated with antiaging effects are quercetin and resveratrol which have now been identified NR4A1 ligands. This is now further discussed in the review (lines     ).

  • There is a typographical error: “Pulomary” should be corrected to “Pulmonary.”

Response: Correction is made.

  • The presented data from mouse and cell models do not directly prove that NR4A1 protects against aging.

Response: As indicated above the data does not prove that NR4A1 protects against aging however the results suggest that NR4A1 may contribute to protection from aging. This is based on extensive animal model studies showing that NR4A1 knockout mice have a shorter lifespan than wild-type mice, the loss of NR4A1 enhances stressor-induced tissue damage and NR4A1 ligands enhance the protective effects of NR4A1. Moreover, the identification of an increasing number of polyphenolics as NR4A1 ligands strengthens the case.

  • The assumption that the health-promoting effects of flavonoids are due to NR4A1 activity is speculative and does not account for the many other known biological targets of these compounds.

Response: I have now added a sentence at the end of section 5.3 indicating that these results suggest that NR4A1 may play a role in some of the flavonoid-mediated responses associated with the beneficial health effects of these compounds (lines 612-620).

Round 2

Reviewer 3 Report

Comments and Suggestions for Authors

All remarks have been added.